# Preliminary Exploration of Economic Polypropylene-Fiber-Reinforced ECC with Superfine River Sand (SSPP-ECC) Applied to a Bridge Pavement Leveling Overlay

**DOI:** 10.3390/ma15072474

**Published:** 2022-03-27

**Authors:** Feihong Wan, Zhiqing Zhu, Wensheng Wang, Guojin Tan, Runchao Yang, Zhicong Zhang

**Affiliations:** 1College of Transportation, Jilin University, Changchun 130022, China; wanfh1719@mails.jlu.edu.cn (F.W.); zhuzq19@mails.jlu.edu.cn (Z.Z.); wangws@jlu.edu.cn (W.W.); zhangzc1719@mails.jlu.edu.cn (Z.Z.); 2Highway Administration Bureau of Jilin Province, Changchun 130022, China; ww934959@163.com

**Keywords:** bridge pavement, leveling overlay, ECC, interface bonding, restrained shrinkage

## Abstract

In response to the current common disease of concrete leveling overlays of bridge pavement in China, the feasibility of using an economic SSPP-ECC with local waste superfine sand as an alternative material for a leveling overlay was proposed in this study. To evaluate the interface bonding property in the girder between the SSPP-ECC and concrete, a slant shear test and split tensile test were designed to study the interfacial shear and tensile properties of the ordinary concrete/ordinary concrete (OC/OC) and ordinary concrete/SSPP-ECC (OC/ECC), where the results showed that SSPP-ECC could significantly improve the interface shear stress and split tensile strength compared to ordinary concrete. Furthermore, the damage status of OC/ECC no longer involved fracturing along the interface; instead, each of the two substrates was partially destroyed, which revealed that OC/ECC had a high bonding effect. Moreover, a restrained shrinkage test was carried out to evaluate the shrinkage property of SSPP-ECC, where the result showed that the shrinkage strain of SSPP-ECC was slightly lower than concrete, where the average cracking time for SSPP-ECC was far longer than for ordinary concrete under the same ambient drying conditions; furthermore, the stress rate for SSPP-ECC revealed that it was a low-cracking-risk material. Meanwhile, the crack width of SSPP-ECC was only 0.1 mm after 35 d, which showed that SSPP-ECC had a more substantial crack width control capacity relative to concrete. The test results initially verified the feasibility and great potential of economic SSPP-ECC applied in a bridge pavement leveling overlay.

## 1. Introduction

Bridge pavement is a typical multi-layer structure with different materials. The conventional structure of current bridge pavement consists of a composite asphalt surface layer, a binding layer, and a waterproof leveling layer [1,2,3]. Due to their tremendous differences in the properties of each layer, the interface between the layers of bridge pavement is vulnerable to large stresses and strains under the impact of complex traffic loads and environmental factors, which leads to a series of diseases [4]. In China, the material for leveling overlay in most concrete bridges is usually cast concrete with an average thickness of about 8–10 cm, which adjusts the leveling of the bridge deck, combines with all girders, and plays a specific role in waterproofing. Therefore, this layer is called the leveling overlay in construction. However, current research in bridge pavement focuses on flexible materials that allow for large deformation, such as the composite asphalt surface layer and the binding layer. In contrast, as a transition layer between the rigid layer and the flexible layer, the leveling overlay has not attracted sufficient attention [5,6].

The leveling overlay in bridge pavement is similar to the concrete overlay used for bridge deck repair and reinforcement methods for the laying position and thickness [7]. As the top concrete structural layer in bridge pavement, the overlay can positively contribute to the overall load-bearing capacity and stiffness of the bridge [8]. High-performance materials, such as polymer-modified concrete, steel-fiber-reinforced concrete, and ready-mixed aggregate concrete, are often used in the overlay to enhance the layer’s properties for better performance [9]. However, the general design and construction standards for bridges in China still do not have precise requirements for leveling overlays [10,11]. In practice, materials used in the leveling overlay are generally in the strength grade of C40–C50 concrete, which is consistent with some local Chinese standards [12,13]. Moreover, the newly mixed material used in a leveling overlay needs to be guaranteed to meet the essential construction workability. However, the authors’ research on the pavement damage of concrete bridges in Jilin Province in recent years showed that the transition layer between the bridge deck and the girder using concrete as a leveling overlay material produces various degrees of damage. The causes can be divided into three categories: (I) Concrete used in the leveling overlay lacks cracking resistance. Specifically, concrete is easy to crack under overloading due to complex traffic and significant temperature variations [14,15] or producing early shrinkage cracks due to weak resistance in shrinkage deformation [16]. It is difficult to inhibit the development of cracks in concrete once cracked. The connection between the cracking areas and the girder is significantly reduced and may even lead to reflective cracks and other diseases in the composite asphalt layer. (II) The thickness of the local area in the leveling overlay is too thin. As the girder of precast concrete sometimes appears to have a sizeable prestressing counter-arch, the thickness of these areas cannot reach the design value when casting them. In practical engineering, the thinnest area is only approximately 3 cm, while the maximum coarse aggregate used in concrete is also up to 2–3 cm, which is quite unreasonable. On the one hand, the thinner areas dry faster and are more likely to produce early shrinkage cracks. On the other hand, these areas have more significant local stress under significant temperature variations and overloading effects. Hence, the leveling overlay in this area is most likely to crumble early and lose its function, even affecting the entire set of layer structures [16,17,18]. (III) Poor bonding property between the leveling overlay and girder. Under complex traffic or other adverse loads, the interface between the two layers is damaged by tensile and shear stresses, and in particular, it is more likely to crack near the wet joint junction due to excessive local shear and tensile stress [19,20]. It can be concluded that a more functional, finer aggregate size, and more robust bond of the new material need to be researched to replace the conventional concrete used in the leveling overlay, which is conducive to improving resistance to diseases in bridge pavement.

Since ECC was invented in the 1990s, scholars have carried out several studies on its properties, and it is still one of the popular cementitious materials [21,22,23]. ECC has been widely recognized for its high toughness, structural deformation properties, and crack width control capacity [24,25], which meet the needs of the problem mentioned earlier in (I). Meanwhile, ECC is made of cementitious binders, superfine aggregates with an average particle size of 110 μm, fibers, and water, and the total size of the mixture is much smaller than conventional concrete [21,26,27], which also meets the needs of the problem mentioned earlier in (II). Some studies showed that the bonding between ECC and different types of concrete is more satisfactory. For example, Pakravan [28] studied the bonding property of ECC to concrete by tensile test combined with microscopic analysis. The result showed different types of ECC enhance the bonding property with concrete. The enhancement efficiency depended on the fiber type and the fiber admixture. Likewise, to improve the interfacial bonding strength, some scholars put forward the idea that increasing the interfacial roughness was the most straightforward and effective solution. Tian et al. [29] compared the effects of the construction method, compressive strength, PVA fiber type, and interfacial roughness on the tensile property of ECC/concrete, where the results showed that the influence of interfacial roughness far outweighed other factors. Furthermore, the interfacial shear and tensile properties were the most critical for evaluating the interfacial bonding effect [30,31]. Gao et al. [32] studied the bonding properties of ECC/concrete in different temperatures using slant shear and split tensile tests. The results showed that the interface bonding property of ECC/concrete was more potent than concrete/concrete at the same temperature. The shear stress and split tensile strength of ECC/concrete increased with increasing temperature. Sahmaran et al. [33] carried out an interfacial slant shear test between ECC/concrete and micro-silica concrete/concrete. The result showed that the interfacial bonding property of ECC/concrete was significantly higher. Meanwhile, ECC was often applied to reinforce a variety of FRP structures. The ECC layer showed promising effects regarding inhibiting slip and considerably improving the carrying capacity of FRP under different loads, such as bending, tension, and shearing [34,35,36,37]. Hence, ECC generally showed a desirable bonding effect when combined with various concrete structures, which, in theory, meets the needs of the problem mentioned earlier in (III). However, a high proportion of cementitious binders were used in the ECC mixing with superfine silica sand aggregates and the PVA or PE fibers’ prices were relatively high [23,38,39], leading to a much higher cost than concrete. The economics issue has become the main resistance to the widespread application in ECC [40,41]. In response to this situation, Tan et al. [42] developed a green and economical SSPP-ECC using local waste superfine river sand, fly ash, and domestic short PP fibers, and carried out a complete study of its recommended mix proportion and essential properties, verifying that the SSPP-ECC had a much higher deformation and crack width control capacity than concrete through a four-point bending test. Tan et al. [43] used the acoustic emission technique to delineate the unique cracking characteristics of SSPP-ECC, and through economic analysis, revealed that SSPP-ECC had more financial and engineering practicality than conventional ECC. In addition, Tan et al. [44] used fractal theory to quantitatively evaluate the characteristic of the whole cracking process in SSPP-ECC and developed a prediction model. Overall, SSPP-ECC is a promising and economical new material, but its specific engineering application research is yet to be carried out.

In summary, taking SSPP-ECC as a material in bridge pavement may provide better effects. A series of studies were carried out to explore the feasibility of SSPP-ECC used in leveling overlays. First, the basic mechanical properties of SSPP-ECC were tested to evaluate the feasibility of applying it to the leveling overlay. Second, the interfacial bonding of the shear stress and the split tensile strength were analyzed to evaluate the bonding properties of SSPP-ECC with ordinary concrete. Finally, considering the fact that the casting area of the leveling overlay covered the whole bridge deck and the proportion of cementitious binders of ECC was high, a restrained shrinkage test was carried out to study the shrinkage properties of SSPP-ECC during the hardening stage. Thus, the tests in this study were undertaken to explore the feasibility of SSPP-ECC as a new material for leveling overlays and to verify its critical application properties, which have profound research significance. It is hoped to minimize the occurrence of diseases associated with leveling overlays and to contribute to the application of functional cementitious composite material in bridge pavements.

## 2. Materials and Methods

### 2.1. Raw Materials

The constituent materials of SSPP-ECC were all fine sized, i.e., cementitious binders, PP fiber 30 μm in diameter and 12 mm in length, local superfine river sand with a maximum particle size of 0.6 mm, and water, which obviously has better adaptability to some thin pouring condition in the leveling overlay than ordinary concrete with coarse aggregates. The cementitious binders included the ordinary Portland cement 52.5 R and F.II grade fly ash produced by Jilin Yatai Group (Jilin, China), whose components and properties are shown in Table 1. The short PP fiber produced by Kaitai Co (Shanghai, China) was used as reinforcement, whose properties are shown in Table 2. The superfine river sand was from the edge of the Yitong river (Jilin, China), with a fineness modulus of 1.1, a maximum particle size of 0.6 mm, an average particle size of 0.25 mm, and mud content of 0.95%. Figure 1a illustrates the grading curve of the superfine sand used in the SSPP-ECC. The ordinary concrete with strength grade C50 was also prepared as a comparison; Figure 1b illustrates the grading curve of the sand and coarse aggregate used in the concrete. Meanwhile, the polycarboxylate-based high-range water reducer (HRWR) produced by Shanghai Chenqi Co (Shanghai, China). with about 1.2% cement quality was used to control the fluidity in the slump of 160–180 mm. 

### 2.2. Specimen Preparations

The mix proportion of SSPP-ECC was determined based on the existing research in the group and ordinary concrete was formulated with C50 as the target [43,44]. The mixture proportions of SSPP-ECC and concrete are shown in Table 3. 

The preparation processes for the SSPP-ECC were as follows: first, cement, fly ash, and some of the water (to give a water–cement ratio of 0.26) were added while mixing slowly for 1 min; second, PP fiber and HRWR were added with slow mixing for 1 min and then mixed quickly for 2 min. Finally, superfine river sand and the remaining water were added with slow mixing for 1 min and then mixed quickly for 3 min to complete the mixing, as shown in Figure 2a. Ordinary concrete was produced using a concrete mixer (manufacturer, city, state, country) according to the specifications shown in Figure 2b [45]. Finally, the mixtures were cast into test specimen molds and demolded after 24 h. According to the specifications, the specimens were transferred to a curing room and cured for 28 d under standard conditions (relative humidity greater than 95%, temperature with 20 ± 2 °C) to complete the specimen preparation.

### 2.3. Test Program

#### 2.3.1. Basic Mechanical Properties

To compare with the concrete that is currently widely used in leveling overlays, the basic mechanical properties of SSPP-ECC were tested first. According to the Chinese specification JTG 3420—2020 [45], the compressive test, Young’s modulus test, and flexural test were conducted. The 150 mm × 150 mm × 150 mm cube specimens were used for the compressive test by loading at a rate of 0.3 kN/s with reference to JTG 3420—2020 Section 5.2 T 0553—2005, as shown in Figure 3a. Young’s modulus was measured by using 150 mm × 150 mm × 300 mm prismatic specimens by loading at a rate of 0.6 MPa/s with reference to JTG 3420—2020 Section 5.2 T 0556—2005, as shown in Figure 3b. The 100 mm × 100 mm × 400 mm specimens were prepared for the flexural test by loading at a rate of 0.06 MPa/s with reference to JTG 3420—2020 Section 5.2 T 0558—2005, as shown in Figure 3c. Meanwhile, after the specimens were prepared, each specimen was cured for 28 d after being transferred into the curing room, and each group was guaranteed to have the tests repeated three times.

#### 2.3.2. Slant Shear Test

For the bonding interface between the leveling overlay and the girder, the main damage is caused by shear or tensile stresses. The slant shear test was carried out to evaluate the interfacial shear resistance property between SSPP-ECC and ordinary concrete. The specific specimen size is shown in Figure 4a. Fabrication methods were as follows: first, ordinary concrete was produced in a specified shape. Meanwhile, the effect of the interface treatment on the shear resistance was considered, where the surface of the ordinary concrete was set to be untreated or scratched; for scratch treatment specimens, to ensure that the influence of the scratch treatment was as identical as possible, the depth, width, and spacing were strictly limited, the depth of the scratches was controlled at 0.5 cm, the width of the scratches were controlled at 0.5 cm, and the spacing between adjacent scratches were guaranteed to be nearly 3 cm, as shown in Figure 4b. Then, the specimens were completed by casting SSPP-ECC or ordinary concrete in the other half, as shown in Figure 4c. The specimens were cured for 28 d, and each group was guaranteed to be repeated three times. Then, the slant shear test was carried out, as shown in Figure 4d, with reference to the specification [46], with loading at a rate of 0.3 kN/s until the specimen was damaged; the load–deformation curve of the entire process, the peak damage load, damage status, and the interface bonding effect were recorded. Finally, the shear stress was calculated according to Equation (1):*τ_n_* = *P* sin*α* cos*α*/*A*(1)
where:*τ_n_*—shear stress (MPa);*P*—maximum applied compressive force (kN);*A*—cross-section angle (mm^2^);*α*—interface angle (30°).

#### 2.3.3. Split Tensile Test

The split tensile test was carried out to evaluate the interfacial tensile property between SSPP-ECC and ordinary concrete. The specific specimen size is shown in Figure 5a. The fabrication methods were as follows: First, ordinary concrete was produced in a specified shape and the effect of the interface treatment on the tensile resistance was also considered; the surface of the ordinary concrete was set to be untreated or scratched, where the specific manufacturing method of scratches remained the same as in Section 2.3.2., as shown in Figure 5b. Then, the specimen was completed by casting SSPP-ECC or ordinary concrete in the other half, as shown in Figure 5c. The specimens were cured for 28 d, and each group was guaranteed to be repeated three times. Then, the split tensile test was started, as shown in Figure 5d, where the specimen was loaded at a rate of 0.1 kN/s until the specimen was damaged [45]. The load–deformation curve of the whole process, the peak damage load, the damage status, and the interface bonding effect were recorded. Finally, the split tensile strength was calculated according to Equation (2):*f_s_* = (2 *P*)/(π *A*)(2)
where:*f_s_*—split tensile strength (MPa);*P*—maximum applied load (kN);*A*—area of the bonded load (mm^2^).

#### 2.3.4. Restrained Shrinkage

Under conventional service conditions, cracks in bridge decks are caused by temperature changes, shrinkage in concrete, etc. Among these factors, shrinkage in concrete had the most significant impact on the development of cracks [46,47]. Considering that when casting SSPP-ECC as the leveling overlay, the entire bridge deck is covered, which involves a large layer of cast-in-place cementitious structure, it is necessary to study its shrinkage and cracking resistance properties during the hardening stage. In this study, a restrained shrinkage test was carried out to evaluate the shrinkage cracking resistance of SSPP-ECC and ordinary concrete with reference to ASTM C 1581-04. The geometry of the specimen is shown in Figure 6a. The inner and outer molds were composed of steel, the strain gauges were symmetrically attached to the inner surface of the ring at the middle height, and the two strain gauges were adhered at 180° to each other. Before the test, SSPP-ECC and ordinary concrete were cast into the molds. Then, the specimens were moved into the test environment within 10 min; after that, the strain gauges were attached to the DH3818Y strain measurement within two minutes. Immediately, the upper surface of the specimen was coated with wet burlap and polyethylene file. After curing for 24 h, the outer ring was removed, and the upper surface was coated with paraffin wax to ensure that the specimens lost water only through the sides. The initial time and strain value were recorded and the curing environment (temperature of 21 ± 2 °C and relative humidity of 50 ± 2%) was set with reference to ASTM C 1581-04; the strain value caused by the shrinkage of the specimen was measured every 15 min. It is usually considered that a sudden decrease in strain of shrinkage strain–time curve is often accompanied by cracking moment of the test specimens [48,49]. All specimens were monitored for at least 28 d.

According to ASTM C 1581-04, the net cracking age *t_cr_* and the average stress rate *S* were used to evaluate the cracking resistance. The cracking age was easily obtained from a moment of an abrupt change in the strain–time curve. The crack number and crack distribution were recorded, and the crack observation instrument was used to measure the crack width, as shown in Figure 6b. Based on the test results, the net strain *ε_net_* was calculated as the difference between the strain in the steel ring at each recorded time and the initial time. The elapsed time *t* was calculated as the difference between each recorded time and the initial time. Plotting the net strain against the square root of the elapsed time for each strain gauge on the test specimen and using linear regression analysis to fit a straight line through the data. The strain rate factor was the slope of the line, according to Equation (3):*ε_net_* = *α t*^1/2^ + *k*(3)
where:*ε_net_*—net strain (m/m);α—strain rate factor for each strain gauge on the test specimen ((m/m)/d^1/2^);*t*—elapsed time (d);*k*—regression constant.

After obtaining the rate factor *α* from each strain gauge, these values were transformed into the average rate factor |*α_avg_*| of two strain gauges and the stress rate *q* was calculated according to Equation (4):*q* = *G* |α*_avg_*|/(2 *t_r_*^1/2^)(4)
where:*q*—stress rate in each test specimen (MPa/d);*G* = 10.47 × 10^6^ psi (72.2 GPa);|α*_avg_*|—absolute value of the average strain rate factor for each test specimen ((m/m)/d^1/2^);*t_r_*—elapsed time at cracking or elapsed time when the test was terminated (d).

Finally, the average stress rate for the test specimens was calculated with an accuracy of 0.01 MPa/d. According to the value of *S*, the shrinkage cracking risks of the SSPP-ECC and ordinary concrete were evaluated.

## 3. Results and Discussions

### 3.1. Basic Mechanical Properties Test Results

The results of the comparison of the basic mechanical properties between ordinary concrete and SSPP-ECC are shown in Figure 7. It can be seen that the compressive strength of SSPP-ECC was slightly lower than ordinary concrete, while it still met the basic requirements in the leveling overlay [12]. Young’s modulus of SSPP-ECC was 25.17 GPa, which was 37.3% lower than ordinary concrete with 34.57 GPa, meaning that SSPP-ECC could withstand more significant deformation. As is known, the asphalt mixture used for bridge deck pavement is viscoelastic [50,51], while the girder is usually rigid concrete. For a given deformation, a leveling overlay with a lower Young’s modulus has lower stress, resulting in lower cracking potential [52], which reveals the potential of SSPP-ECC as an alternative for concrete overlay sandwiched between asphalt pavement and concrete girder. The flexural strength of SSPP-ECC was 6.59 MPa, which was 45.5% higher than ordinary concrete with 4.53 MPa, meaning it had a stronger ability in tensile resistance. Based on the above mechanical properties test results, SSPP-ECC met the property requirements for the leveling overlay in construction and had better bending, tensile resistance, and deformation capacity.

### 3.2. Slant Shear Test

The slant shear test reflects the interfacial shear resistance; the resulting load–deformation curves are shown in Figure 8. It was found that with the growth in deformation, the load of all specimens first increased slowly, then rapid growth to the peak occurred, and finally, it dropped abruptly with the test’s end, which indicated that the failure damage of all interfaces was brittle and occurred instantaneously. It was also evident that there were obvious differences between the interfacial shear resistance of each group, i.e., ordinary concrete/ordinary concrete without scratch treatment (OC/OC-N) < ordinary concrete/ordinary concrete with scratch treatment (OC/OC-Y) ≈ ordinary concrete/SSPP-ECC without scratch treatment (OC/ECC-N) < ordinary concrete/SSPP-ECC with scratch treatment (OC/ECC-Y). The average results of the shear stress are shown in Figure 9. The shear stress of OC/OC-N was 5.28 MPa, which increased to 8.01 MPa after the scratching treatment, with an improvement of 51.78%. Meanwhile, the shear stress of OC/ECC-N was 8.91 MPa, which increased to 14.16 MPa after the scratching treatment, with an improvement of 58.88%. The results could be analyzed from two aspects. On the one hand, the scratch treatment had a strong enhancement effect on the interfacial shear stress, and the enhancement effect of SSPP-ECC was slightly higher than ordinary concrete. On the other hand, the interfacial shear stress of ordinary concrete/SSPP-ECC (OC/ECC) was generally higher than ordinary concrete/ordinary concrete (OC/OC), with an improvement effect from 68.75% to 76.63%; relative to OC/OC, OC/ECC had better shearing properties.

During the loading process of the slant shear test, the absorption energy was determined using the area under the load–deformation curve shown in Figure 8. The average absorption energy calculation results are shown in Figure 10. Similar to the shear stress results, OC/ECC absorbed more energy than OC/OC. Meanwhile, the scratch treatment had a significant enhancement in the improvement of absorption energy. The reason can be found in combination with Figure 8. On the one hand, the specimens with the scratch treatment were subjected to a higher load; on the other hand, they experienced more deformation during the loading process. Therefore, they were able to absorb more energy.

Figure 11 shows the shear damage status of each group. Figure 11a shows that the damage of OC/OC-N was directly separated by excessive slippage along the bonding interface, which was caused by the relatively too smooth interface. Although OC/OC-Y had significantly better shear stress, the damage still occurred at the bonding interface, and the specimens were divided into two complete parts, accompanied by the interfacial scratches that were filled with ordinary concrete; the specimen was damaged because it was sheared off at the scratches, as shown in Figure 11b. Figure 11c shows that the damage status of OC/ECC-N was similar to OC/OC-N; the shear damage also occurred at the interface, where it can be found that a small part of SSPP-ECC fell off and adhered to the ordinary concrete after the fracture. The above groups of tests were basically similar to the previous related studies in that the specimens were damaged along the bonding interface [53]. However, the damage status of OC/ECC-Y was very disparate from the other groups, as shown in Figure 11d, where the outer edge of the fracture surface was damaged along the bonding interface, while the inner part was no longer wholly developed along the bonding interface. It can be seen that a large area of SSPP-ECC inside the fracture surface was sheared off and firmly adhered to the ordinary concrete matrix, while the interfacial scratches were not damaged. It is interesting to note that, due to greater interfacial bond strength, longitudinal cracks resulting from the loading already appeared on the surface of the ordinary concrete before the specimen reached the edge of shear stress, as shown in Figure 12a. Since OC/ECC-Y had the highest shear stress, its concrete matrix even showed complete longitudinal cracks after the end of loading, as shown in Figure 12b, which indicated that the shear resistance of ordinary concrete/SSPP-ECC was very good.

### 3.3. Split Tensile Test

The split test reflects the interfacial tensile property. The load–deformation curves are shown in Figure 13. It was found that, with the growth of deformation, the load on all specimens increased slowly at first, then there was rapid growth to the peak, and finally, the interface underwent brittle fracture and the test ended immediately. Figure 13 shows that there was a noticeable difference between the tensile resistances in each group, i.e., OC/OC-N < OC/OC-Y < OC/ECC-N < OC/ECC-Y. The average results of the split tensile strength are shown in Figure 14, where the split tensile strength of OC/OC-N was 2.12 MPa, which increased to 2.75 MPa after the scratching treatment, with an improvement of 29.72%. Meanwhile, the split tensile strength of OC/ECC-N was 3.42 MPa, which increased to 4.44 MPa after the scratching treatment, with an improvement of 29.82%. It was found that the upgrading effect of OC/ECC was the same as OC/OC. Compared with the significant improvement effect of the scratch treatment on the shear resistance property in Section 3.2., the scratch treatment had a negligible effect on improving the tensile resistance property. On the other hand, the split tensile strength of OC/ECC was generally higher than that of OC/OC. The enhancement effect was from 61.23% to 61.45%; OC/ECC had a better tensile resistance property than OC/OC.

During the loading process of the split tensile test, the absorption energy was determined by the area under the load–deformation curves shown in Figure 11. The average absorption energy calculation results are shown in Figure 15. Similar to the split tensile strength results, OC/ECC generally absorbed more energy than OC/OC. Meanwhile, the interfacial absorption energy in the specimens with the scratch treatment was higher than the untreated ones because the scratch treatment was subjected to a higher load. However, compared with the absorption energy in Section 3.2, the specimens with the scratch treatment had lower absorption energy in the split tensile test because all the specimens were subjected to similar deformation, while in the slant shear test, the specimens were subjected to significantly different loading deformations, as shown in Figure 8 and Figure 13.

Figure 16 shows the interfacial damage status after the split tensile test. Figure 16a shows that the damage status of OC/OC-N developed entirely along the interface; only a tiny amount of later cast ordinary concrete was spalled off at the edge of the interface, which was caused by the relatively weak bond. Although the split tensile strength was improved for the OC/OC-Y specimen, as shown in Figure 16b, the split damage status still occurred at the interface; however, due to excessive compressive stress, the edges of the specimen showed an uneven rupture surface, and the root of the scratch at the bonding interface was wholly pulled off. Figure 16c showed that the damage status of OC/ECC-N also developed along the bonding interface, but about half of the area of SSPP-ECC was pulled off and dislodged into the ordinary concrete part. The above groups of tests were basically similar to the previous related studies in that the specimens were damaged along the bonding interface [31]. However, the split damage status of OC/ECC-Y was different from other groups, as shown in Figure 16d. Both the SSPP-ECC and ordinary concrete matrixes had obvious damage fractures, and the fracture surfaces no longer wholly developed along the bonding interface. It can be seen that part of the ordinary concrete was detached from the whole specimen, and the other part was still combined with the SSPP-ECC matrix, which indicated that the tensile effect of the OC/ECC bonding interface was significantly more potent than that of OC/OC.

### 3.4. Restrained Shrinkage

Figure 17 shows the strain variation on the inner surface of the steel ring due to shrinkage during the hydration reaction stage for SSPP-ECC and concrete. In the first 2 d, the shrinkage strains for the two materials showed a rapid growth trend, causing an approximately 60 × 10^−6^ growth strain before both entered a low-rate growth phase. It can be seen that the shrinkage strain growth rates of the two materials had apparent differences, where the concrete’s properties led to the strain growing faster. The strain–time curve changed abruptly at about 7 d, where noticeable cracks could be observed on the surface of the specimen, as shown in Figure 18a. In contrast, SSPP-ECC went into a long shrinkage process, and only slight microcracks were observed at 24 d and 30 d, as shown in Figure 18b. Compared with concrete, SSPP-ECC was prepared from more homogeneous and finer aggregates. The internally dispersed fibers exerted a significant tensile stress resistance capacity, causing specimens to be less prone to cracking and having a small crack width.

The statistical results of the maximum strain and the cracking time in the steel ring by the two materials are shown in Figure 19. It was found that the average maximum strain in the restrained shrinkage of the concrete was 100.10 × 10^−6^, while the average maximum strain of SSPP-ECC was 86.85 × 10^−6^, indicating that the strain of SSPP-ECC was slightly lower than that of concrete. Although SSPP-ECC used fewer aggregates, the high fly ash content significantly slowed the volumetric shrinkage in the hydration stage. Meanwhile, the bond strength between the fibers and matrix gradually grew. The resistance to shrinkage cracking was steadily enhanced; therefore, the shrinkage cracking time of SSPP-ECC was much longer than that of concrete. It was concluded that SSPP-ECC had better volume stability.

To compare the shrinkage crack development conditions of the two materials, the crack width of each group was measured using a crack observation instrument every 7 d. The average crack width results are shown in Figure 20. The initial crack width of concrete already reached 0.5–0.6 mm at 7 d. Since then, the width had increased over time. The average crack width of the parallel specimens exceeded 1 mm by 28 d, while the SSPP-ECC specimens cracked at 24 and 30 d, with initial crack widths of 0.04 and 0.05 mm, respectively. The average crack width of the parallel specimens did not exceed 0.1 mm by 35 d, which confirmed the crack width of SSPP-ECC at 35 d was much smaller than the initial crack width of the concrete specimens, which also showed that SSPP-ECC had a high crack width control capacity. On the other hand, as shown in Figure 7, the flexural strength of SSPP-ECC was significantly higher than that of concrete. Fibers greatly contributed to the matrix’s tensile property, which led to SSPP-ECC specimens being hard to crack in the restrain shrinkage test.

As shown in Table 4, the average stress rate *S* was calculated for each group according to Equations (3) to (4). The cracking risk can be evaluated according to ASTM C 1581-04. The results showed that the average cracking time for concrete is 6.5 d, while the average stress rate *S* was 0.595, which meant that concrete was a high-cracking-risk material. In contrast, the average cracking time for SSPP-ECC was 27 d, which was not above 28 d to qualify as low cracking risk. While its average stress rate *S* was only 0.073, it was sufficient to meet the requirements of low cracking risk for stress rate such that SSPP-ECC can be considered a low cracking risk material.

The strain value measured on the surface of the inner ring in the restrain shrinkage test is theoretically related to the actual shrinkage deformation and tensile stress [54]. Referring to the theoretical equations in [54], the shrinkage deformation and the average tensile stress on the outer surface of the specimen can be calculated as shown in Figure 21 and Equations (5) to (7).
*q* = −[1 − (*r*_1_/*r*_2_)^2^] *E*_2_
*ε_θ_*(5)
*σ_θ_* = (2 *r*_2_^2^)/(*r*_3_^2^ − *r*_2_^2^) *q*(6)
*q* = (−*ε_sh_*)/(1/*E*_1_ × ((1 − *μ*_1_) *r*_2_^2^ + (1 + *μ*_1_) *r*_3_^2^)/(*r*_3_^2^ − *r*_2_^2^) + 1/*E*_2_ × ((1 − *μ*_2_) *r*_1_^2^ + (1 + *μ*_2_) *r*_2_^2^)/(*r*_2_^2^ − *r*_1_^2^))(7)
where:*q*—interface stress;*E_1_*—Young’s modulus of ordinary concrete/SSPP-ECC;*E_2_*—Young’s modulus of the steel ring;*μ_1_*—Poisson’s ratio of ordinary concrete/SSPP-ECC;*μ_2_*—steel ring Poisson’s ratio;*σ_θ_*—outer ring tensile stress;*ε_θ_*—inner ring shrinkage strain;*ε_sh_*—outer ring shrinkage strain.

The average tensile stress on the outer surface of the specimens can be obtained from the association of Equations (5) and (6), and the shrinkage deformation on the outer surface of the specimens can be obtained from the association of Equations (5) and (7). Since the concrete specimens already reached the maximum shrinkage strain and cracked after 7 d, while the SSPP-ECC specimens had the maximum shrinkage strain after approximately 27 d, the average tensile stress and shrinkage deformation on the outer surface of the concrete at 7 d and the SSPP-ECC at 7 d and 27 d are shown in Table 5.

As shown in Table 5, when calculating the shrinkage deformation at the outer surface, the concrete specimens produced a strain of 642 × 10^−6^ at 7 d, while the SSPP-ECC was slightly lower than concrete with 618 × 10^−6^. After that, the concrete specimens showed larger shrinkage cracks, while SSPP-ECC specimens continued to resist shrinkage and the shrinkage strain only increased by 14% after 27 d. More importantly, the maximum tensile stress at the outer surface of the concrete specimens at 7 d was 11.930 MPa, while in SSPP-ECC, the tensile stress was 9.061 MPa, and even the maximum tensile stress of SSPP-ECC specimens after the shrinkage time reached 27 d was lower than the maximum tensile stress in concrete at 7 d, indicating that the SSPP-ECC had stronger resistance to shrinkage and cracking. The tensile stress of SSPP-ECC shrinkage was more minor; therefore, the problem of excessive local stress caused by shrinkage is less likely to occur when casting SSPP-ECC in leveling overlay.

## 4. Conclusions

To evaluate the feasibility of economic SSPP-ECC as an alternative material for bridge pavement leveling overlay, the authors conducted an experimental analysis of its interfacial bonding effect and resistance to shrinkage cracking. The main conclusions were as follows:

The slant shear test results showed that interfacial scratch treatment provided a better enhancement effect on split tensile strength and using SSPP-ECC instead of ordinary concrete in shear stress provided more significant enhancement than a scratch treatment. Moreover, the damage status of OC/ECC-Y was no longer completely damaged along the bonding interface as in other specimens, which showed that OC/ECC-Y had an excellent shear resistance property.The split tensile test results showed that the interfacial scratch treatment provided an enhancement effect regarding split tensile strength, while the lifting effect was low relative to the shear stress; using SSPP-ECC instead of ordinary concrete provided a more significant enhancement in split tensile strength than a scratch treatment. The damage status of OC/ECC-Y was no longer showed complete damage along the bonding interface as in other specimens, which showed that OC/ECC-Y had excellent tensile resistance.The restrained shrinkage test showed that the average cracking age of SSPP-ECC was far longer than ordinary concrete, which showed that SSPP-ECC could significantly delay the cracking time. The shrinkage strain of SSPP-ECC was slightly lower than concrete, and its average stress rate confirmed SSPP-ECC as a low-cracking-risk material. Moreover, the crack width of SSPP-ECC was far lower than ordinary concrete, which confirmed SSPP-ECC had a powerful crack width control capacity.In summary, SSPP-ECC had a proficient bonding effect with ordinary concrete, which had stronger resistance in shear and tension. SSPP-ECC had better volume stability and substantial shrinkage cracking resistance. Thus, SSPP-ECC can be applied to the construction of a large area and low thickness of the leveling overlay in bridge pavement.

In this study, the feasibility of economical SSPP-ECC as a leveling overlay material for bridge pavement was investigated through a laboratory test, which initially revealed that SSPP-ECC had a stronger interfacial property and shrinkage cracking resistance compared to ordinary concrete. More work will be completed in the future, such as conducting a detailed numerical simulation analysis, fatigue test analysis under impact loading, and more practical engineering studies. These are the main contents that the author will carry out in subsequent experimental designs.

## Figures and Tables

**Figure 1 materials-15-02474-f001:**
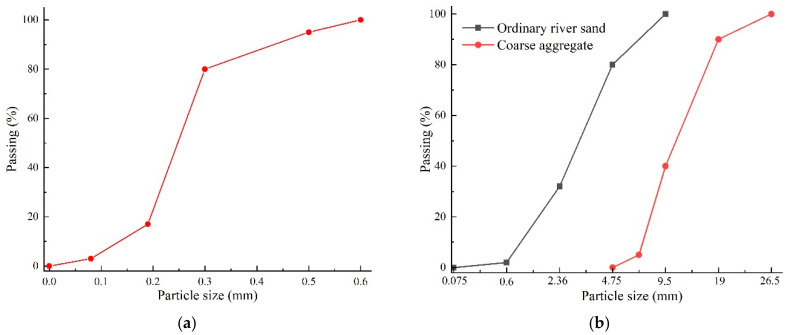
Grading curve. (**a**) Superfine river sand for the SSPP-ECC; (**b**) Sand and coarse aggregate for ordinary concrete.

**Figure 2 materials-15-02474-f002:**
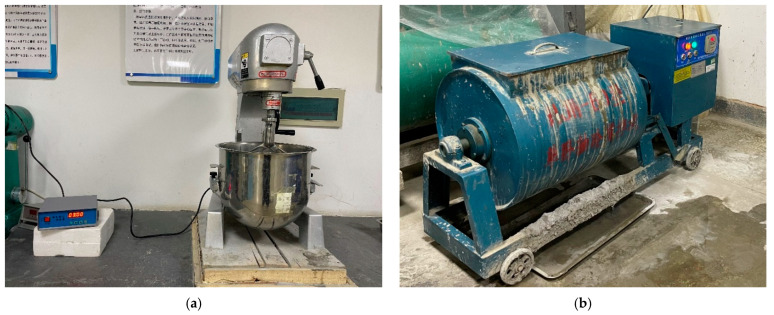
Specimen preparations. (**a**) SSPP-ECC preparation; (**b**) Concrete preparation.

**Figure 3 materials-15-02474-f003:**

Basic mechanical tests. (**a**) Compression test; (**b**) Young’s modulus test; (**c**) Flexural test.

**Figure 4 materials-15-02474-f004:**
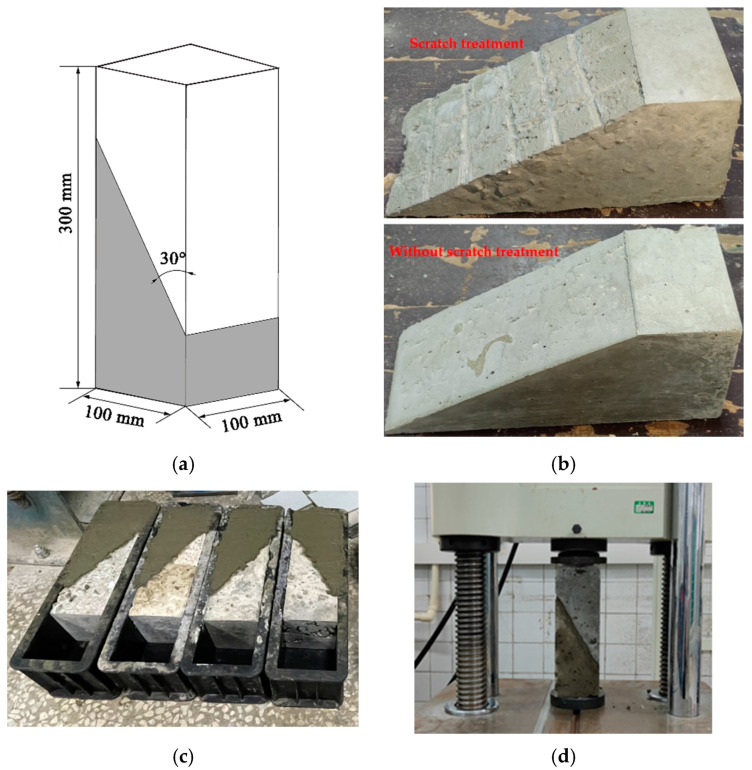
The slant shear test flow. (**a**) Specimen geometry; (**b**) Concrete interfacial scratching treatment; (**c**) Casting; (**d**) Loading.

**Figure 5 materials-15-02474-f005:**
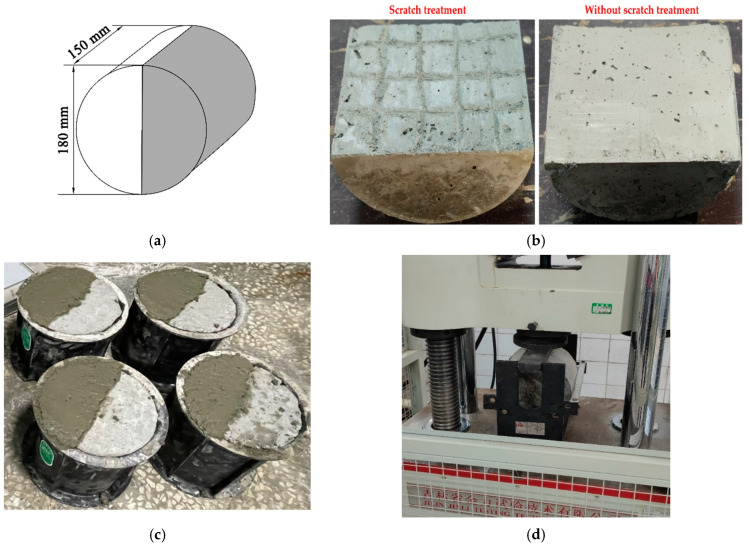
The split tensile test flow. (**a)** Specimen geometry; (**b**) Concrete interfacial scratching treatment; (**c**) Casting; (**d**) Loading.

**Figure 6 materials-15-02474-f006:**
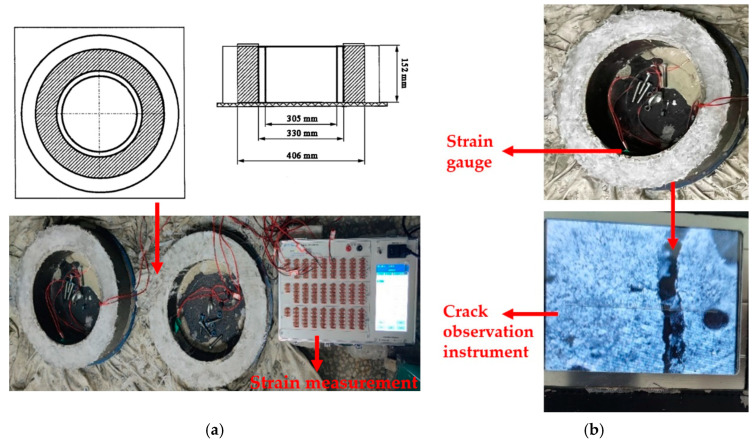
Restrained shrinkage flow. (**a**) Specimen preparation; (**b**) Crack width measurement.

**Figure 7 materials-15-02474-f007:**
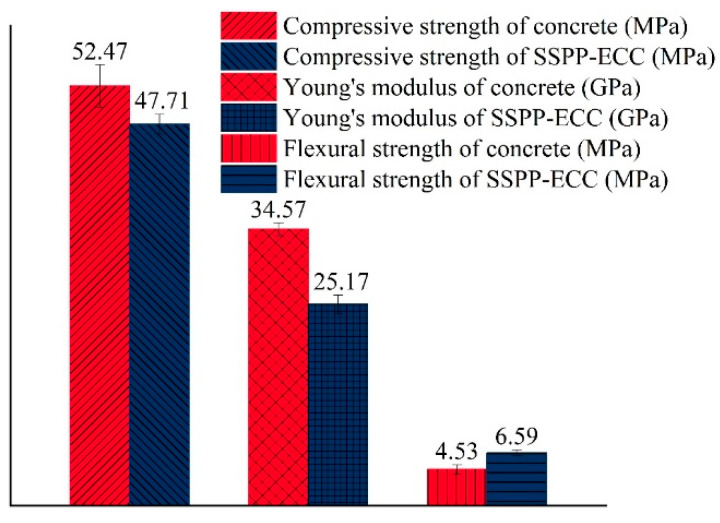
Results of the basic mechanical properties test.

**Figure 8 materials-15-02474-f008:**
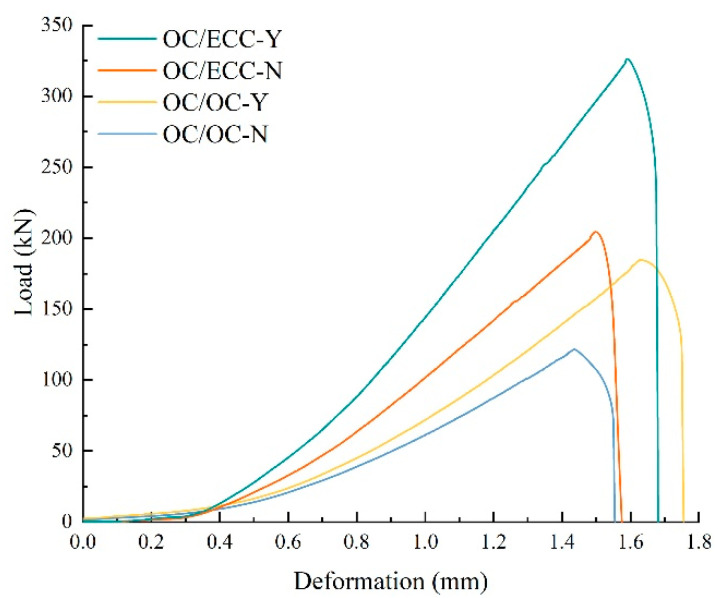
Load–deformation curve from the slant shear test.

**Figure 9 materials-15-02474-f009:**
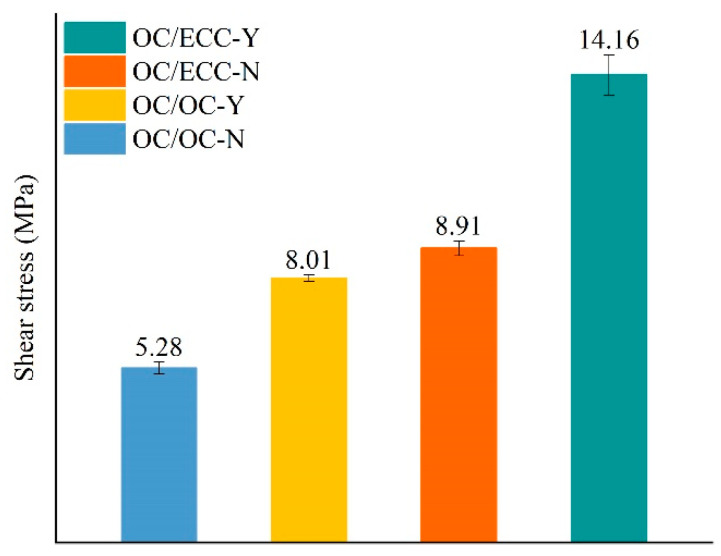
Shear stress of each group.

**Figure 10 materials-15-02474-f010:**
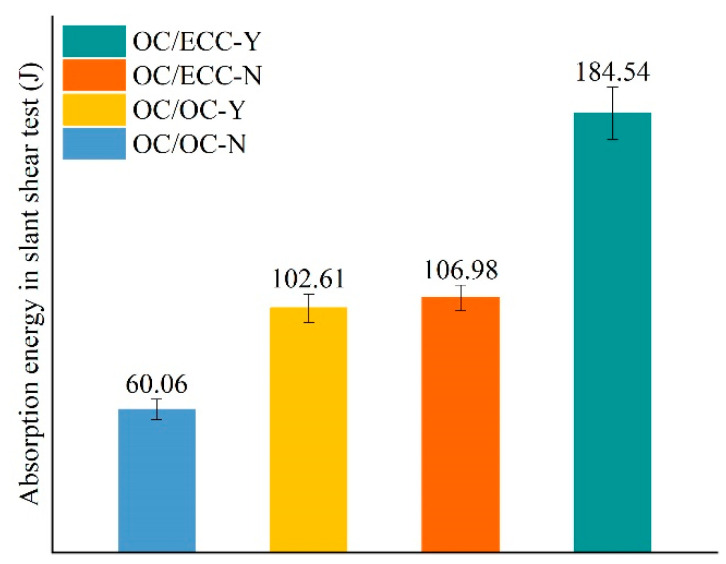
Absorption energy of each group in the slant shear test.

**Figure 11 materials-15-02474-f011:**
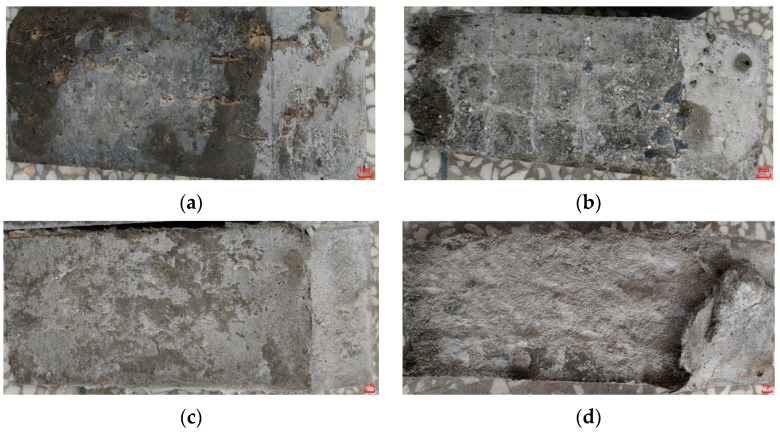
Interface damage status of the slant shear test specimens. (**a**) OC/OC-N; (**b**) OC/OC-Y; (**c**) OC/ECC-N; (**d**) OC/ECC-Y.

**Figure 12 materials-15-02474-f012:**
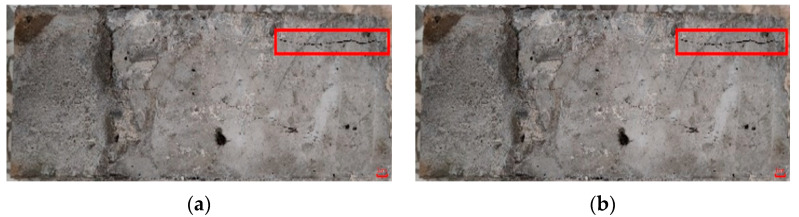
Partial cracking damage status of the OC/ECC slant shear test specimens. (**a**) OC/ECC-N; (**b**) OC/ECC-Y.

**Figure 13 materials-15-02474-f013:**
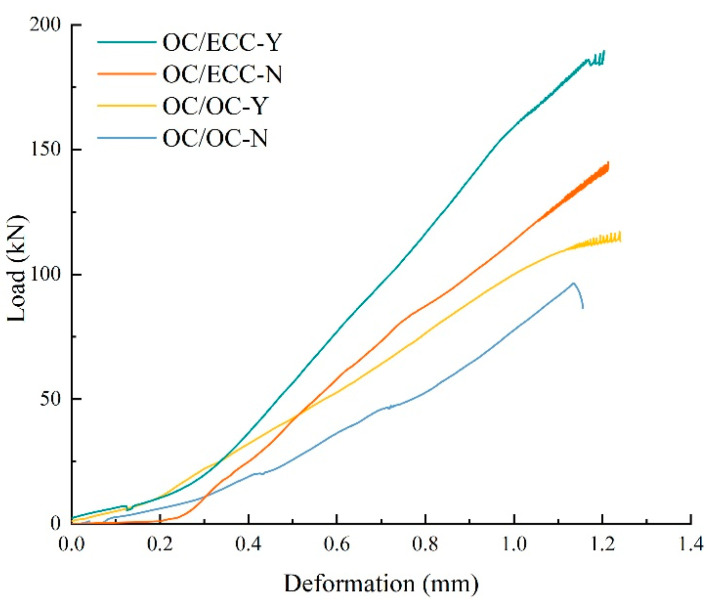
Load–deformation curve of the split tensile test.

**Figure 14 materials-15-02474-f014:**
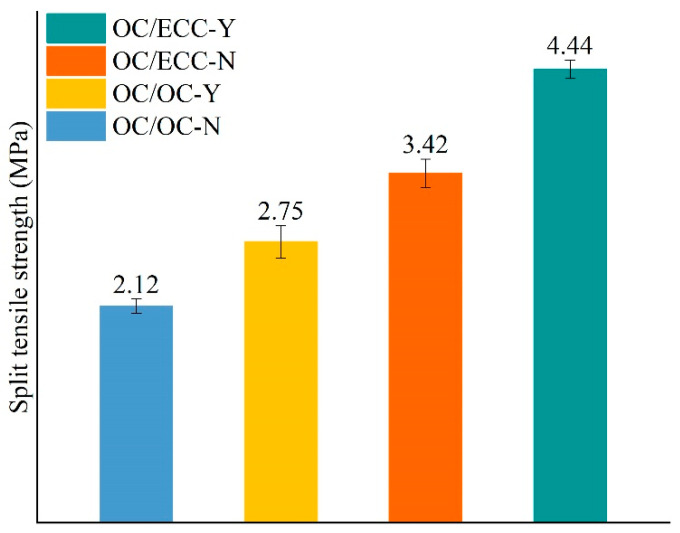
Split tensile strength of each group.

**Figure 15 materials-15-02474-f015:**
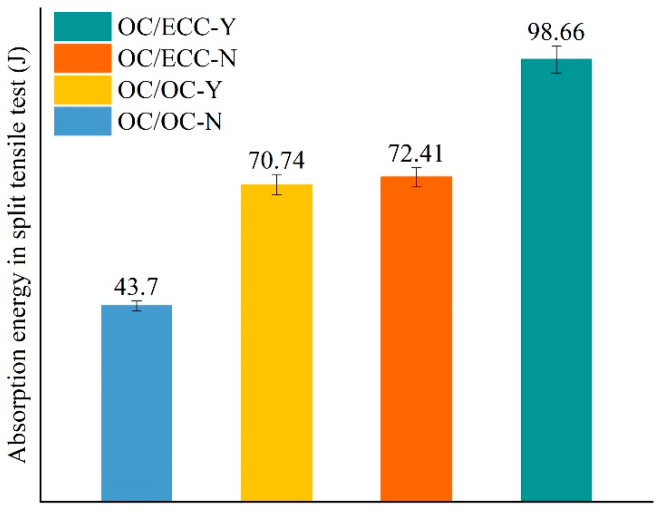
Absorption energy in the split tensile test of each group.

**Figure 16 materials-15-02474-f016:**
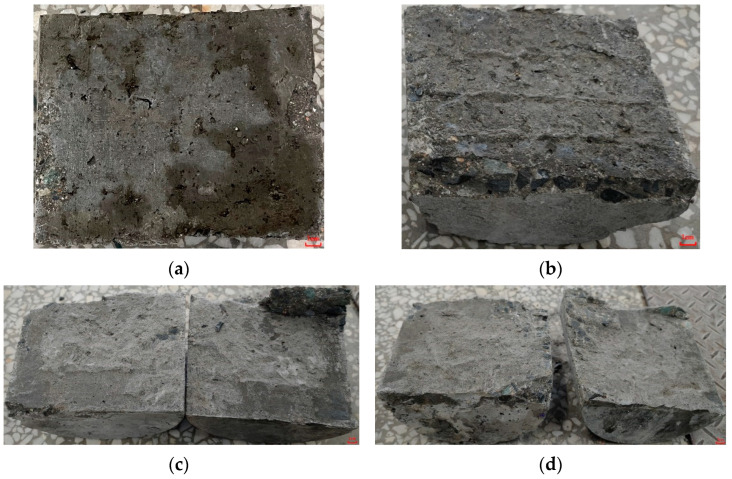
Interface damage status of the split tensile test specimens. (**a**) OC/OC-N; (**b**) OC/OC-Y; (**c**) OC/ECC-N; (**d**) OC/ECC-Y.

**Figure 17 materials-15-02474-f017:**
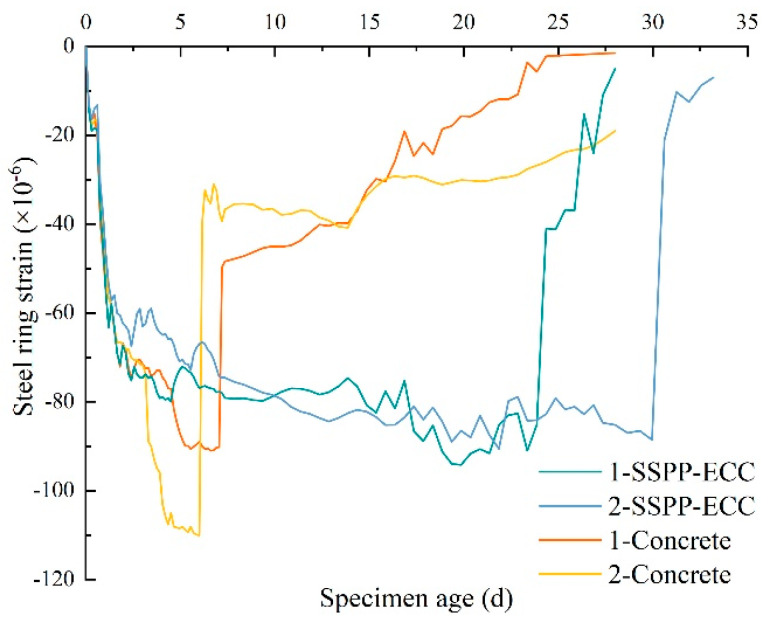
Shrinkage strain–time curve.

**Figure 18 materials-15-02474-f018:**
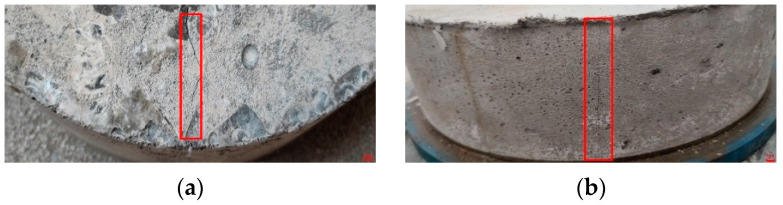
Crack status. (**a**) Concrete; (**b**) SSPP-ECC.

**Figure 19 materials-15-02474-f019:**
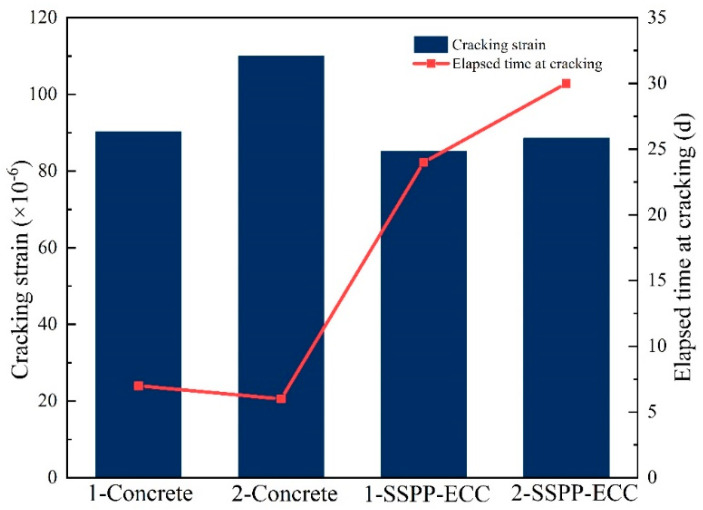
Cracking strain and cracking time of concrete and SSPP-ECC.

**Figure 20 materials-15-02474-f020:**
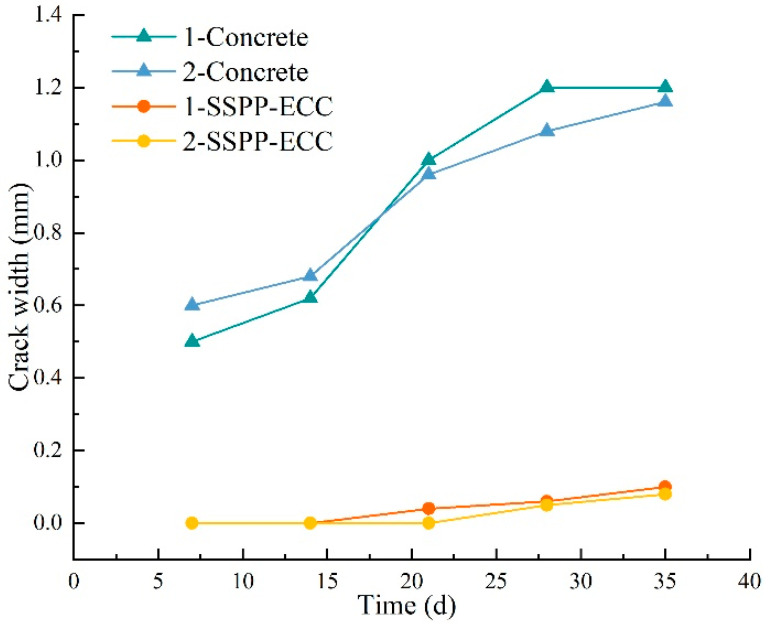
Crack width of concrete and SSPP-ECC at different times.

**Figure 21 materials-15-02474-f021:**
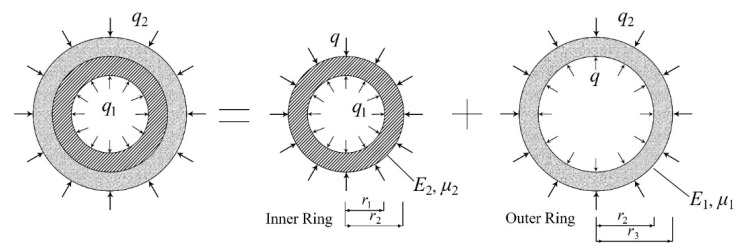
Simplified graph of the stress in the steel ring.

**Table 1 materials-15-02474-t001:** Main components and properties of binders.

Properties	Cement	Fly Ash
Specific gravity	3.10	2.13
Surface area ratio (m^2^/kg)	370	420
CaO (%)	60.38	3.01
SiO_2_ (%)	21.11	50.37
Al_2_O_3_ (%)	6.04	27.62
Fe_2_O_3_ (%)	2.56	7.83
MgO (%)	1.08	1.85
Loss on ignition (%)	1.02	7.23
Water ratio (%)	0.11	0.81

**Table 2 materials-15-02474-t002:** Physical properties of PP fiber.

Diameter (μm)	Length (mm)	Density (g/cm^3^)	Young’s Modulus (GPa)	Nominal Strength (MPa)	Elongation at Break (%)
30	12	0.91	3.5	500	20

**Table 3 materials-15-02474-t003:** Mix proportions of SSPP-ECC and Concrete.

Type	Mix Proportion (kg/m^3^)							
	Cement	Fly Ash	Superfine River Sand	Ordinary River Sand	Coarse Aggregate	PP Fiber	Water	HRWR
Concrete	445	——	——	630	1220	——	150	7
SSPP-ECC	520	520	520	——	——	18.2	312	10

**Table 4 materials-15-02474-t004:** Potential for cracking classification.

*t_cr_* (d)	*S* (MPa/d)	Potential for Cracking	Concrete		SSPP-ECC	
0 < *t_cr_* ≤ 7	*S* ≥ 0.34	High	*t_cr_*	*S*	*t_cr_*	*S*
7 < *t_cr_* ≤ 14	0.17 ≤ *S* < 0.34	Moderate–high	6.5	0.595	27	0.073
14 < *t_cr_* ≤ 28	0.10 ≤ *S* < 0.17	Moderate–low				
*t_cr_* > 28	*S* < 0.10	Low				

**Table 5 materials-15-02474-t005:** The tensile stress and shrinkage deformation on the outer surface of the specimens.

Type	Time (d)	*ε_θ_* (×10^−6^)	*q* (MPa)	*σ_θ_* (MPa)	*ε_sh_* (×10^−6^)
Concrete	7	100.100	3.064	11.930	642
SSPP-ECC	7	76.015	2.327	9.061	618
	27	86.850	2.658	10.350	706

## Data Availability

The data presented in this study are available on request from the corresponding author.

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
