# Peer review of "Preliminary Exploration of Economic Polypropylene-Fiber-Reinforced ECC with Superfine River Sand (SSPP-ECC) Applied to a Bridge Pavement Leveling Overlay"

_materials, 2022, doi:10.3390/ma15072474_

Round 1
Reviewer 1 Report
This paper presents evaluation the bridge pavement leveling overlay. For this purpose, current common disease of concrete leveling overlay of bridge pavement in China were studied. Comparisons between the results shows that the feasibility and great potential of economic SSPP-ECC applied in bridge pavement leveling overlay.
This is an interesting paper. However, this reviewer does not recommend the publication of the manuscript in the present form because of the following reasons:
1) This reviewer think it will be useful if the authors provide some additional information on the Materials and Methods used in this study.
2) English usage and spelling should be improved. Some parts seem to be copied and pasted from PDF and needs attention.
3) The manuscript is focused on properties of complex pavement and needs better description of the properties of materials, such as linear and nonlinear viscoelastic properties. They may use available literature such as the following reference:
- (2020). Effect of Evotherm-M1 on Properties of Asphaltic Materials Used at NAPMRC Testing Facility. Journal of Testing and Evaluation, 48(3).
- (2019). Characterization and validation of the nonlinear viscoelastic-viscoplastic with hardening-relaxation constitutive relationship for asphalt mixtures. Construction and Building Materials, 216, 648-660.
This paper will recommend for publication if the authors consider the above suggestions to improve the quality of the manuscript. Some editing is still needed. I believe it will be done before publishing.
Reviewer 2 Report
Please address to the following issues:
- review the article for the correction of symbols - you often forget to use italics when writing variables either in the text and equations, e.g. line 244, 270, Eq.1, Eq.2, etc.
- correct subscripts across the text
- please correct units, it should be kN not KN, and kN/s instead of KN/S?
- line 179 correct “text”
- line 257, “(m/m)/d1/2” is this correct?
- what is the objectivity of sample preparation regarding to scratching treatment. i.e. the result may obviously depend on scratching treatment which is not uniform and random
- listing very specific numbers in conclusion is inappropriate since only very limited number of samples were tested. You cannot stated that SPP-ECC is better by specific percentage than OC after testing one sample in each test. Please reformulate conclusions. You should rather state some basic thoughts and preliminary insights than exact numbers.
- the issue of possible stochastic errors has been completely ignored – please put some comment on this
Reviewer 3 Report
General Comment
The manuscript presents an experimental study on the mechanical performance of polypropylene fiber reinforced ECC with superfine river sand (SSPP-ECC) to be applied for bridge pavement levelling overlay. For this, SSPP-ECC was produced in laboratory, as well as ordinary concrete for comparison. A series of tests were carried out to explore the feasibility of SSPP-ECC and its potential to be used for levelling overlay, namely: compressive tests, slant shear tests, split tensile tests and restrained shrinkage. The manuscript describes the used methodology, the raw materials, the specimens’ preparation for testing, the test program and the testing procedures. The results are presented and discussed. From the obtained results, the authors conclude about the feasibility and great potential of economic SSPP-ECC to be applied in bridges pavement levelling overlay.
The topic of the manuscript is very interesting since it deals with novel functional and economic cementitious based materials used for levelling overlay to improve bridge pavement resistance during service life. Engineering application research of SSPP-ECC, as a promising and economical material, is still need. In particular, the application of this material for bridge pavement levelling overlay is novel.
I made few comments in order to improve the manuscript. The authors should take the comments into account and revise their manuscript.
Specific Comment 1
Please revise the entire manuscript to improve the reading and correct some typos.
Specific Comment 2
Section 3
Throughout Section 3, please briefly discuss how your findings compare with the ones from previous related studies. This would improve the quality of your manuscript.
Reviewer 4 Report
The article covers the topic of the Preliminary exploration of economic polypropylene fiber reinforced ECC with superfine river sand (SSPP-ECC) applied to bridge pavement leveling overlay.The subject and the supporting knowledge are informative and present added value to the body of knowledge on the subject area. The manuscript has very good cohesion.
I suggest that the following minor modification should be considered to improve the quality of paper:
1. Abstract need to be rewritten to report about the main and new findings obtained in this paper briefly.
2. I suggest to add separated point - Research significance - Please descibe here the main essence of the research.
Why presented studies are so important?
3. Table 2 - "elongation" - I suggest change to "elongation at the break".
4. Table 3 - please change "Kg" to "kg". Moreover please add size of grains.
5. I suggest to avoid to call a SSPP-ECC as concrete, this is mortar (based on the content).
6. Line 176 - please change "150 m×150 mm×150 mm" to "150 mm×150 mm×150 mm. Please improve similar mistake in lines 178 and 180.
7. Please determine why this speed rates were choosen? ( for 0.3 KN/s and 0.06MPa/s).
8. Line 201 - please change "KN" to "kN" in whole manuscript.
9. Figures 16 and 18 - please add a scale.
Round 2
Reviewer 1 Report
Great effort has been done by the authors, and the paper can be considered for publication.